# Bio-SELEX: A Strategy for Biomarkers Isolation Directly from Biological Samples

**DOI:** 10.3390/mps6060109

**Published:** 2023-11-11

**Authors:** Juan David Ospina-Villa, Valentina Restrepo-Cano, Miryan Margot Sánchez-Jiménez

**Affiliations:** Instituto Colombiano de Medicina Tropical-ICMT, Universidad CES, Sabaneta 055450, Colombia; restrepoc.valentina@uces.edu.co (V.R.-C.); msanchez@ces.edu.co (M.M.S.-J.)

**Keywords:** Bio-SELEX, aptamers, biomarkers, biological samples, pull down, mass spectrometry

## Abstract

Bio-SELEX is a revolutionary method for the discovery of novel biomarkers within biological samples, offering profound insights into diagnosing both infectious and non-infectious diseases. This innovative strategy involves three crucial steps: Traditional SELEX, Pull Down, and mass spectrometry. Firstly, Traditional SELEX involves the systematic selection of specific nucleic acid sequences (aptamers) that bind to the target molecules of interest. These aptamers are generated through iterative rounds of selection, amplification, and enrichment, ultimately yielding highly selective ligands. Secondly, the Pull-Down phase employs these aptamers to capture and isolate the target biomarkers from complex biological samples. This step ensures the specificity of the selected aptamers in binding to their intended targets. Lastly, mass spectrometry is utilized to identify and quantify the captured biomarkers, providing precise information about their presence and concentration in the sample. These quantitative data are invaluable in disease diagnosis and monitoring. Bio-SELEX’s significance lies in its ability to discover biomarkers for a wide range of diseases, spanning infectious and non-infectious conditions. This approach holds great promise for early disease detection, personalized medicine, and the development of targeted therapies. By harnessing the power of aptamers and mass spectrometry, Bio-SELEX advances our understanding of disease biology and opens new avenues for improved healthcare.

## 1. Introduction

The SELEX strategy was discovered in 1990 by two groups of independent researchers [1,2]. This technique makes it possible to obtain aptamers, which are ssDNA and ssRNA molecules, which adopt unique three-dimensional (3D) structures allowing them to recognize specific targets with high affinity and specificity.

Since then, the creation of multiple variations in the SELEX technique has evolved. Negative-SELEX [3], Counter-SELEX [4], Capillary electrophoresis SELEX (CE-SELEX) [5], Microfluidic-SELEX [6], and Cell-SELEX [7] are some of the variants that, based on advances in molecular biology, biomedical engineering, and biotechnology have improved the time of acquisition and the ability to recognition of the aptamers by different types of targets.

Our research group has been developing a variant of the SELEX strategy that can be adaptable to multiple studies. Therefore, we report a new SELEX strategy modification to identify new biomarkers from biological samples called Bio-SELEX. The word Bio refers to the search for biomarkers and the biological nature of the samples used to obtain them.

In the domain of biomedical research, the pursuit of effective biomarkers facilitating early disease diagnosis and guiding tailored therapeutic interventions is a persistent objective.

Biomarkers are fundamental molecules in biomedical research, as they allow progress in the discovery of new drugs, treatment, or monitoring the state of a disease. The identification of biomarkers is essential, but the methods for their discovery are still limited. Aptamers (single-stranded DNA or RNA) have been proven to be a powerful tool for biomarker discovery. Several researchers have employed aptamers to discover biomarkers in various diseases such as lung cancer [8], leukemias [9], ovarian cancer [10,11], and Chagas disease [12], among many others. However, it is evident that the SELEX strategies utilized in these studies exhibit notable diversity, highlighting the absence of a standardized protocol for biomarker discovery from biological samples. This lack of a uniform approach for biomarker identification presents a significant challenge in biomedical research, underscoring the need to establish guidelines or a common framework to advance more effectively in this field. Within this context, Bio-SELEX emerges as a transformative methodology poised to advance biomarker discovery within biological samples, transcending the boundaries of infectious and non-infectious diseases. This innovative strategy encompasses three fundamental phases: Traditional SELEX, Pull-Down, and mass spectrometry.

The crux of Bio-SELEX resides in Traditional SELEX, a meticulously orchestrated process involving the systematic selection of nucleic acid sequences known as aptamers. These aptamers are iteratively generated through rounds of selection, amplification, and enrichment, demonstrating a remarkable ability to selectively bind to target molecules of interest. This foundational step serves as the cornerstone for subsequent biomarker identification.

After Traditional SELEX, the Pull-Down phase is implemented to harness the specificity of the aptamers. This phase facilitates the capture and isolation of target biomarkers from complex biological samples, ensuring precise and selective interactions between aptamers and their intended molecular targets. Ultimately, the culmination of Bio-SELEX is realized through mass spectrometry, a technique offering invaluable insights into biomarker identification and quantification. By furnishing precise information regarding the presence and concentration of captured biomarkers in the sample, mass spectrometry assumes a pivotal role in disease diagnosis and monitoring.

The extensive applicability of Bio-SELEX spans a spectrum of diseases, encompassing both infectious and non-infectious conditions. Beyond promising early disease detection, this method lays the groundwork for personalized medicine and the development of targeted therapeutic strategies. By harnessing the synergistic capabilities of aptamers and mass spectrometry, Bio-SELEX is poised to deepen our understanding of disease biology and facilitate enhanced healthcare approaches. In this context, we explore the many facets of Bio-SELEX, illuminating its transformative potential in the arena of biomarker discovery and clinical application. Notably, this protocol has already demonstrated its efficacy in uncovering novel biomarkers for Chagas disease, with ATPase alpha emerging as a promising biomarker. It provides a valuable means of detecting the chronic disease phase, acknowledged as the most challenging stage for parasite detection [13].

## 2. Experimental Design

The Bio-SELEX strategy (Figure 1) is employed to identify biomarkers within diverse biological samples containing proteins. It is crucial to obtain control samples from “healthy” individuals, as they are essential for identifying differentially expressed proteins when compared to the test samples. To ensure an accurate representation of protein expression within the study group, it is recommended to create sample pools for each evaluated group. Various methods for extracting proteins from biological samples are available. In our study, we utilized a straightforward and cost-effective approach, specifically the Trizol method. Following protein extraction, quantification was performed using commercially available methods (we employed the BCA Protein Assay Kit ab102536 from Abcam). Subsequently, protein integrity was assessed via SDS-PAGE electrophoresis at 12%.

Depending on the nature of the biological sample, the removal of abundant proteins such as albumin from serum may be required. Several commercial products based on affinity, such as Cibacron blue, are available to facilitate the removal of albumin from biological samples. The extracted proteins are then categorized into control and experimental groups.


**SELEX Strategy**


We executed the SELEX strategy following the protocol published by Wang et al. in 2019 [14] with some modifications. An ssDNA library of 80 nucleotides was utilized, consisting of a 40-nucleotide variable region (N) flanked by two conserved regions (5’GTCTATATGATCTAACTC(40N)CCAGCAGTGAGTCATCAGAT-3’) at both ends to aid with detection and amplification. A forward oligonucleotide complementary to the 3’ conserved region (5’-GTCTATATGATCTAACTC-3’) and a reverse oligonucleotide complementary to the 5’ conserved region (5’-ATCTGATGACTCACTGCTGG-3’) were employed. The library and oligonucleotides were synthesized by Macrogen USA (Rockville, MD, USA) The library is amplified using 2X PCR Hot Start Master Mix with dye (ABM). The number of amplification cycles has to be standardized, typically ranging from 10 to 40 cycles, to identify the optimal cycle count without the presence of parasite DNA. Subsequently, 0.2 nM of the library is used for subsequent rounds.


**Negative Selection**


Proteins extracted from control groups were immobilized onto 96-well plates using carbonate/bicarbonate buffer and allowed to become fixed via incubation overnight at 4 °C (between 1 µg/mL and 400 µg/mL in a volume of 100 µL). Following immobilization, the plates underwent three washes with PBS 1X, followed by the addition of 100 µL of the library at a concentration of 0.2 nM in SELEX 1X buffer. The interaction between the library and the immobilized proteins occurred for 1 h at temperatures between 18 and 24 °C with agitation at 100 rpm. Un-bound molecules were subsequently recovered, amplified via PCR, and served as the starting point for positive selection rounds. This process was repeated for 3 to 4 rounds.


**Positive Selection**


Proteins extracted from experimental groups were fixed to 96-well plates using carbonate/bicarbonate buffer (12.5 mM Sodium bicarbonate, 87.5 mM Sodium carbonate, pH 9.6) overnight at 4 °C (between 1 µg/mL and 400 µg/mL in a volume of 100 µL). After fixation, three washes with PBS 1X were conducted, followed by the addition of 100 µL of the library selected after negative rounds at a concentration of 0.2 nM in SELEX 1X buffer. The interaction between the library and the immobilized proteins took place for 1 h at temperatures between 18 and 24 °C with agitation at 100 rpm. Un-bound molecules were discarded, and the bound molecules were recovered and amplified via PCR. This process was repeated for 10 to 20 rounds.


**Next Generation Sequencing (NGS)**


The final library obtained after initial rounds of negative and positive selection was prepared and subsequently subjected to sequencing using Illumina 2 × 250 equipment.


**Data Analysis (NGS)**


Sequencing results were processed using Cutadapt software 4.6 [15] to eliminate adaptor sequences and conserved regions from the 3’ and 5’ ends of libraries, retaining only the 40 nt random regions. The quality of obtained sequences was evaluated using FASTQC software [16]. FASTAptamer software v1.0.3 [17] was employed to perform various functions, including tallying, standardization, evaluation, and organization of individual sequences within groups. It also analyzed and compared the distribution of sequences between different populations, grouped sequences into families based on their similarity using the Levenshtein edit distance, determined the degree of enrichment for all sequences across various populations, and extensively searched for recurring nucleotide sequence patterns.


**Aptamer Synthesis**


Once aptamers were selected based on their representativeness, the 40 nucleotide (nt) aptamers were synthesized on a scale of 200 nanomoles (nmoles) and purified by Polyacrylamide gel electrophoresis (PAGE) and coupled to biotin at the 5’. These sequences were ordered from Macrogen USA (Rockville, MD, USA).


**Biomarker Isolation (Pull-Down)**


Aptamers coupled to biotin were mixed with magnetic beads coupled to streptavidin to form the Aptamer–biotin/streptavidin–magnetic beads complex. Subsequently, biological samples (1 mL of pure or diluted complete sample) were added. The incubation time should be standardized as it depends on the strength of the interaction between the selected aptamer and the target protein. This can vary between one or several hours at temperatures around 18–24 °C or overnight at 4 °C in moderate agitation of 100 rpm [18]. The complexes of biomarkers/Aptamer–biotin/streptavidin–magnetic beads were isolated using a magnetic rack for 2–5 min. Three washes were performed with Binding buffer 1X, followed by dilution of the obtained pellet with Laemmle buffer 2X.


**Biomarker Identification (LC-MS/MS)**


The resolved complexes from experimental groups and control groups (comprising only magnetic beads, magnetic beads coupled to aptamers without biological samples, and magnetic beads coupled to aptamers exposed to biological samples from control groups) were subjected to SDS-PAGE electrophoresis. Bands observed in the experimental group lines were excised using sterile lancets and stored in 1.5 mL Eppendorf tubes containing 100–200 µL of sterile water. These samples were then sent for identification via liquid chromatography coupled to mass spectrometry (LC-MS/MS) through the services provided by Creative Proteomics (Shirley, NYUSA) using Nanoflow UPLC: Ultimate 3000 nano UHPLC system (ThermoFisher Scientific, Waltham, MA, USA).


**Data Analysis (LC-MS/MS)**


Results obtained from LC-MS/MS were analyzed and compared against protein databases corresponding to the sample’s species and relevant microorganisms using Maxquant software (1.6.2.6). It is important to note that identified biomarkers could include differentially expressed proteins from both the host and microorganisms as responses to infection. In cases of non-infectious diseases, the analysis of differential protein expression between control and experimental groups would be sufficient.


**Experimental Validation**


To validate the interaction between selected aptamers and identified biomarkers in vitro, recombinant proteins were obtained, and aptamers were synthesized. Binding affinity assays were performed, including Electrophoretic Gel Shift Assay (EMSA), Enzyme-linked Aptasorbent Assay (ELASA), isothermal titration calorimetry (ITC), surface plasmon resonance (SPR), and Biolayer Interferometry (BLI), among others, depending on each laboratory’s capacity.

### 2.1. Materials

TRI Reagent (93289, Sigma-Aldrich, Darmstadt, Germany);BCA Protein Assay Kit (ab102536, Abcam, Cambridge, UK);2X PCR Hot Start Master Mix with dye (G900-dye, ABM, Richmond, Canada);GeneJET Gel Extraction Kit (K0691, Thermo Scientific™, Waltham, MA, USA);Dynabeads™ M-270 Streptavidin (65306, Invitrogen™, Waltham, MA, USA);2x Laemmli Sample Buffer (1610737, Bio-Rad, Hercules, CA, USA);InstantBlue^®^ Coomassie Protein Stain (ab119211, Abcam, Cambridge, UK).

### 2.2. Equipment

S1000 Thermal Cycler (BIORAD, Hercules, CA, USA);NanoDrop™ 2000/2000c Spectrophotometers (ND2000, Thermo Scientific™, Waltham, MA, USA);MagneSphere 12-Tube Magnetic Separation Stand (Z5343, Promega, Madison, WIS, USA);Mini-PROTEAN^®^ Tetra Vertical Electrophoresis Cell, 4-gel, for 1.5 mm thick handcast gels (1658006FC, Hercules, CA, USA).

## 3. Procedure

To carry out the SELEX strategy, it is necessary to request a ssDNA library of 80 nucleotides with a variable region (N) of 40 nucleotides flanked by two conserved regions (5′GTCTATATTCTATATGATCTAACTC(40N)CCAGCAGTGAGTCATCAGAT-3′) at the ends. This will facilitate detection and amplification. As well, a forward oligonucleotide complementary to the 3′ conserved region (5′-GTCTATATTCTATATGATCTAACTC-3′) and a reverse oligonucleotide complementary to the 5′ conserved region (5′-ATCTATCTGATGACTCACTGCTGG-3′) are required from your trusted provider (we use Macrogen, Rockville, Maryland, USA).

### 3.1. SELEX Strategy

Prepare a PCR mix using your preferred Taq polymerase [we use 2X PCR Hot Start Master Mix with dye (ABM)] (Table 1 and Table 2).

Purify PCR amplicons using any of the commercially available kits (we use GeneJET Gel Extraction Kit).Heat 200 pmol of amplified and purified library in 500 µL of Binding buffer 1X at 95 °C by 5–10 min, and let it cool slowly to promote 3D structure formation.Fix 100 ng of the extracted proteins in 96-well dishes or polypropylene tubes using 1X carbonate-bicarbonate buffer and incubate at 4 °C overnight.Wash 3–4 times using PBS 1X buffer.Bring the library from step 3 into contact with the fixed proteins from step 5 and allow interaction for 1 h at temperatures between 18 and 24 °C while agitating at 100 rpm.Wash 3–4 times using Binding buffer 1X.Recover protein-bound sequences by adding 100 µL of hot nuclease-free H_2_O (95 °C) by shaking at 100 rpm for 5 min at temperatures between 18 and 24 °C.Use these 100 µL as a template to be amplified via PCR using 2–5 µL in each reaction under the previously described conditions.Repeat the SELEX strategy for 10–20 rounds.

**OPTIONAL STEP**: One way to determine the number of SELEX rounds to perform is to quantify the amount of bound and non-bound molecules using Nanodrop. If you observe a downward trend in non-bound molecules and an upward trend in bound molecules that remain stable, it will indicate that you do not need to continue making selection rounds.
Send the final round of the SELEX strategy for identification through NGS.

**OPTIONAL STEP**: The data from the final round are sufficient to obtain the necessary information for aptamer selection. However, depending on the laboratory’s resources, it is recommended to sequence each of the selection rounds to identify sequences enriched with each round.
Analyze the sequencing data and choose the aptamers based on their representativeness (readings and readings per million) for synthesis coupled to biotin.

### 3.2. Pull-Down

Add 50 µL (0.5 mg) of magnetic beads coupled to streptavidin in 1.5 mL Eppendorf tube.Place the tube in a magnetic rack for 2–5 min until the beads are on one side of the tube. Remove the supernatant carefully and discard it.Add 1 mL binding buffer 1X, vortex, separate magnetic beads into a magnetic rack for 2–5 min, remove the supernatant carefully, and discard it.Dilute the biotinylated aptamer (100 ng) in 300 µL of binding buffer 1X.Add the 300 µL of biotinylated aptamer to the tubes with magnetic beads coupled to streptavidin that were previously washed.Allow interaction between 18 and 24 °C for 1–2 h at 100 rpm or overnight at 4 °C.Separate aptamer–biotin/streptavidin–magnetic beads complexes into magnetic rack for 2–5 min, remove the supernatant carefully, and discard it.Wash three times with binding buffer 1X, separate complexes into magnetic rack for 2–5 min, remove the supernatant carefully and discard it.Add 1 mL of test sample (pure or diluted depending on biological sample).


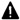
 **CRITICAL STEP:** During the Pull-Down biomarker separation process, negative controls must be performed in parallel to validate the experiment. It is suggested to at least use the following:

1-An experiment where no test sample is used (only 1X binding buffer can be used). This is in order to rule out the presence of proteins that do not come from the sample.

2-An experiment using samples from the control group (1 mL) in order to select the bands differentially expressed in the test group, if observed in both groups.
Allow interaction between 18 and 24 °C for 1–2 h at 100 rpm or overnight at 4 °C.Separate aptamer–biotin/streptavidin–magnetic beads/biomarkers complexes into magnetic rack for 2–5 min, remove the supernatant carefully, and discard it.Wash three times with binding buffer 1X, separate complexes into magnetic rack for 2–5 min, remove the supernatant carefully, and discard it.Resuspend the complexes obtained in 50 µL of 2x Laemmli Sample Buffer.Resolve the complexes obtained for the test group and control groups by electrophoresis in an SDS-PAGE polyacrylamide gel using BIORAD instruments.Stain the proteins present in the gel using Coomassie or silver staining (we use Instablue Coomassie protein stain).


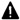
 **CRITICAL STEP:** Verify on the manufacturer’s data sheet that the stain used is compatible with mass spectrometry (MS).
Identify differentially expressed protein bands in the study group.Remove the identified bands using a sterile lancet and place them in a 1.5 mL Eppendorf tube with 100–200 µL of sterile water.Send the bands for protein identification through MS.

### 3.3. LC-MS/MS

The sent samples are processed using In-gel digestion, nano-LC, mass spectrometry, and data analysis, which are sent back for interpretation.

### 3.4. Data Interpretation

The interpretation of the data should be oriented in those proteins that are expressed only in the experimental group and in making a thorough review of the literature to identify their relationship with the disease being studied. In the case of infectious diseases, the biomarker can be both a host protein in response to infection, and a microorganism protein involved in disease pathogenesis. In the case of non-infectious diseases, differential expression of proteins in the study group compared to the control group may be of great importance for understanding the pathogenesis of the disease.

It is important to note that all findings obtained by Bio-SELEX must be confirmed by in vitro experiments to know the recognition capacity of the selected aptamer and the identified biomarker.

## 4. Expected Results

At the end of the SELEX strategy and sequencing the final library, it is expected that a list of aptamers ordered according to their representativeness will be obtained (reads, reads per million), with potential for the recognition of biomarkers. Table 3 shows some problems and solutions that can be presented in the SELEX strategy.

At the end of the **Bio-SELEX strategy,** in its three stages, it is expected to obtain a list of biomarkers with the potential for the diagnosis, treatment, follow-up, or understanding of the disease to be studied. Table 4 shows some problems and solutions that can be presented in the Bio-SELEX strategy.

At the end of the Bio-SELEX strategy, it is expected to obtain a database containing proteins differentially expressed in the test group which can be used as potential biomarkers for a disease and under conditions that can be modified by each researcher according to what is to be studied.

It is important to note that these proteins are identified directly from the biological study sample, which is extremely relevant when setting a future goal that may be for the diagnosis of a disease, follow-up to treatment, or have a better understanding of the pathophysiology of a disease.

## 5. Reagents Setup

**Binding buffer 1X** (PBS containing 2.5 mM of MgCl2, Tween20 to 0.02%, and 1 mM of heparin in miliQ water, pH 7.4).**Carbonate/bicarbonate buffer 1X** (12.5 mM Sodium bicarbonate, 87.5 mM Sodium carbonate, pH 9.6).**PBS 1X buffer** (137 mM NaCl, 2.7 mM KCl, 10 mM Na2HPO4, and 1.8 mM KH2PO4, pH 7.4).

**Limitations:** The Bio-SELEX strategy focuses on the search for biomarkers of protein nature, which may be a limitation for certain pathologies in which molecules of another nature are biomarkers of greater interest.

## Figures and Tables

**Figure 1 mps-06-00109-f001:**
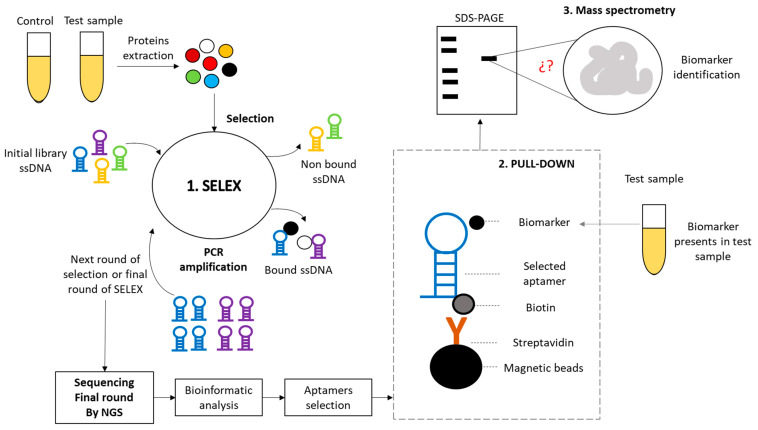
Schematic representation of the methodology needed to identify biomarkers directly from biological samples using the Bio-SELEX strategy.

**Table 1 mps-06-00109-t001:** PCR master mix preparation.

Composition	25 µL	50 µL	Final Conc.
Nuclease-free water	to 25 µL	to 50 µL	
10 µM Forward Primer	95	15–30 s	0.2 μM (0.05~1 µM)
10 µM Reverse Primer	45–68	15–60 s	0.2 μM (0.05~1 µM)
Template library	68	1 kb/min	<1 µg/50 µL
Hot Start Taq 2x PCR Master Mix with Dye	12.5 µL	25 µL	1X

**Table 2 mps-06-00109-t002:** PCR amplification profile of library.

Steps	Temperature (°C)	Time	No. of Cycles
Initial denaturalization	95	5 min	1
Denaturation	95	15–30 s	10–30
Annealing	45–68	15–60 s	
Extension	68	1 kb/min	
Final extension	68	5 min	
Cooling	4	∞	1

**Table 3 mps-06-00109-t003:** SELEX troubleshooting guide.

Problematic Result	Some Possible Causes	Some Potential Solutions
Too many or smear bands after negative or positive SELEX rounds.	Contaminated samples or reagents. Primer concentration to high. Low annealing temperature.	Check controls to determine if problems are with samples or reagents. Run temperature gradient. Run temperature gradient.
No band after negative or positive SELEX rounds.	Too few PCR cycles. Short extension time. High annealing temperature. Degradation of initial library. Inhibitors in sample. Sub-optimal primers.	Optimize primers and cycling parameters. Maintain properly stored library, evaluate protein interaction time and temperature. Review sequence obtained by the provider sequencing.

**Table 4 mps-06-00109-t004:** Bio-SELEX troubleshooting guide.

Problematic Result	Some Possible Causes	Some Potential Solutions
Too many bands after Pull-Down and SDS-PAGE.	Contaminated samples or reagents.	Check controls to determine if problems are with samples or reagents.
No bands observed after Pull-Down and SDS-PAGE.	Low concentration of aptamers. Low concentration of biomarkers in the sample. Insufficient interaction time or inadequate temperature. Insufficient staining of the gels.	Test different concentrations. Test different conditions of interaction. Dye for longer or use a more sensitive method.

## Data Availability

Data is contained within the article.

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
