# Peer review of "Bio-SELEX: A Strategy for Biomarkers Isolation Directly from Biological Samples"

_mps, 2023, doi:10.3390/mps6060109_

Round 1
Reviewer 1 Report
Comments and Suggestions for Authors
In this work Ospina-Villa et al introduce a protocol ‘Bio-SELEX’ which may be used for selecting and enriching aptamer sequences against biomarkers in a ‘test sample’. The method is based on first carrying out a negative selection on extracted and fixed proteins in a control sample and then using the remaining library for positive selection in a ‘positive control’ which may contain a potential protein biomarker. The protocol is described in great detail, and this article would be suitable for publication after the following minor details are addressed:
1) Some additional references should be included in the introduction which lay a foundation for the current work, such as key works related to SELEX which have enabled the current protocol to take shape.
2) Page 3 lines 2 and 12, what is the total concentration of the protein loaded into the 96-well chamber for fixation, would different concentrations impact the BIO-SELEX results?
3) Page 3 line 45, the authors mention that pull down step involves an incubation time of 1 hour, however later, page 7 line 10, the incubation time is mentioned as 1-2 hours or overnight at 4 degrees. What is the appropriate incubation time in this protocol and are there any guidelines which might determine how much incubation might be better.
4) In order to distinguish "differentially expressed protein bands in the study group.", it seems that nontarget-aptamer-based similar pull-down steps should be performed in parallel. The authors should better describe how the negative controls during the pull-down process should be designed.
5) The idea of aptamer-based biomarker discovery and isolation is not completely novel, in addition to their own work (reference 1), the authors should add other reports as well. This is critical to validate the audience that such a method is not successful only in a particular group or study.
6) Some potential challenge or limitation of bio-SELEX method should also be discussed at the end of the manuscript, for future further development.
Comments on the Quality of English Language
Quality of English is good, proofreading is necessary to correct minor grammatical issues.
Author Response
Manuscript ID: mps-2657897
REPLY TO REVIEWER 1
Reviewer(s)' Comments to Author:
Reviewer: 1
Comments and Suggestions for Authors
In this work Ospina-Villa et al introduce a protocol ‘Bio-SELEX’ which may be used for selecting and enriching aptamer sequences against biomarkers in a ‘test sample’. The method is based on first carrying out a negative selection on extracted and fixed proteins in a control sample and then using the remaining library for positive selection in a ‘positive control’ which may contain a potential protein biomarker. The protocol is described in great detail, and this article would be suitable for publication after the following minor details are addressed:
1) Some additional references should be included in the introduction which lay a foundation for the current work, such as key works related to SELEX which have enabled the current protocol to take shape.
Response: Thank you very much for this comment, a more judicious introduction was really needed mentioning the origin of the SELEX strategy. A paragraph was added on page 1, lines 28 to 40, which is underlined in yellow. Likewise, 7 new references (1 to 7) were added which are highlighted in yellow in the section of references.
2) Page 3 lines 2 and 12, what is the total concentration of the protein loaded into the 96-well chamber for fixation, would different concentrations impact the BIO-SELEX results?
Response: The concentration of proteins to be fixed depends on the type of biological sample used, in serum the proteins obtained are abundant, while in other biological samples such as cerebrospinal fluid, and urine, among others a much lower concentration is expected. Each experiment should be standardized using between 1µg/ml to 400 µg/ml of extracted proteins. We used the 100 µg/ml concentration. This information was added to the text and is highlighted in yellow in lines 16-17 and 28.
What allows the Bio-SELEX strategy is to differentiate between the proteins of the control groups and the proteins of the experimental groups. If sufficient protein is not available, the strategy is likely to be unable to identify biomarkers differentially expressed in the experimental groups, so it is recommended to use concentrations > 50 µg/ml if sample availability permits.
3) Page 3 line 45, the authors mention that pull down step involves an incubation time of 1 hour, however later, page 7 line 10, the incubation time is mentioned as 1-2 hours or overnight at 4 degrees. What is the appropriate incubation time in this protocol and are there any guidelines which might determine how much incubation might be better.
Response: Thank you for this observation. In our previous experiments, we used a 1-hour incubation with good results. However, protocols like this I mention below (Arthur Louche, Suzana Pinto Salcedo, Sarah Bigot. Protein-Protein Interactions: Pull-Down Assays. Protein-Protein Interactions: Pull-Down Assays., pp.247-255, 2017, ff10.1007/978-1-4939-7033-9_20ff.ffhal-03012474f) indicate that incubation can take place for an hour, or more, even overnight at 4ºC, and this will depend on how strong the interaction between the obtained aptamer and the target protein is.
We have added a text on page 4 lines 8-11 highlighted in yellow where the incubation time is clarified, and the protocol mentioned above is cited. Similarly, the reference highlighted in yellow is added in the corresponding section with the number 13.
4) In order to distinguish "differentially expressed protein bands in the study group.", it seems that nontarget-aptamer-based similar pull-down steps should be performed in parallel. The authors should better describe how the negative controls during the pull-down process should be designed.
Response: Thank you for this valuable comment. We have added a short description of the 2 negative controls that should be carried out at least on page 7, lines 25 to 30 highlighted in yellow.
5) The idea of aptamer-based biomarker discovery and isolation is not completely novel, in addition to their own work (reference 1), the authors should add other reports as well. This is critical to validate the audience that such a method is not successful only in a particular group or study.
Response: Thank you. We have added a paragraph and several citations to previous works that are on page 2, lines 3 to 14 highlighted in yellow, and added in the references section.
6) Some potential challenge or limitation of bio-SELEX method should also be discussed at the end of the manuscript, for future further development.
Response: We have added a limitation section on page 10, lines 16 to 18 which is highlighted in yellow.
Reviewer 2 Report
Comments and Suggestions for Authors
The paper describes a protocol for the discovery of aptamers for biomarkers present in samples without prior identification of the biomarkers. That in itself is the major query, in a physiological sample there are a large number of proteins present. The authors state that the use of a negative sample cohort will ensure that those proteins not associated with a disease are not detected. However, there is no guarantee that all the proteins present in different, healthy, people are identical. As such the direct discovery of aptamers for proteins not available in the control group does not guarantee that the discovered proteins are actually biomarkers for the disease looked for.
Some minor comments:
- Page 2, line 28, full stop missing after ‘method’
- Throughout the paper, please check the temperature unit, the ° sign appears to be underlined in various places
- Page 6, line 19/20, ‘nonbond’ should be ‘non-bound’
Comments on the Quality of English LanguageThe English in general is fine, some minor comments are provided in the general comments.
Author Response
Manuscript ID: mps-2657897
REPLY TO REVIEWER 2
Reviewer(s)' Comments to Author:
Reviewer: 2
Comments and Suggestions for Authors
The paper describes a protocol for the discovery of aptamers for biomarkers present in samples without prior identification of the biomarkers. That in itself is the major query, in a physiological sample there are a large number of proteins present. The authors state that the use of a negative sample cohort will ensure that those proteins not associated with a disease are not detected. However, there is no guarantee that all the proteins present in different, healthy, people are identical. As such the direct discovery of aptamers for proteins not available in the control group does not guarantee that the discovered proteins are actually biomarkers for the disease looked for.
Thank you very much for your comment. He is absolutely right, that is why it is necessary to use appropriate controls that minimize this possibility and it is also made clear that all findings obtained by Bio-SELEX must be confirmed experimentally.
Some minor comments:
- Page 2, line 28, full stop missing after ‘method’
Response: Thank you, it was corrected in the new version.
- Throughout the paper, please check the temperature unit, the ° sign appears to be underlined in various places
Response: Thank you, it was corrected in the new version.
- Page 6, line 19/20, ‘nonbond’ should be ‘non-bound’
Response: Thank you, it was corrected in the new version.
Reviewer 3 Report
Comments and Suggestions for Authors
I am submitting my revision of the Manuscript ID mps-2657897, entitled “ Bio-SELEX: A strategy for biomarkers isolation directly from 2 biological samples”.
This manuscript presents a novel method, Bio-SELEX, for the discovery of new biomarkers within biological samples. The concept explored in this paper is interesting and merits further investigation. The content aligns with the Journal's scope.
Nevertheless, there are several areas that require improvement before it can be published. Many crucial details are omitted, and the information provided lacks depth. Furthermore, a thorough revision of the procedural section is needed to enhance clarity and comprehensibility.
In general the experimental design need to be reviewed and English improved.
Here are some suggestions for the authors to consider during the revision process:
· Page 3, line 1 to 20:
Kindly elucidate the distinctions between positive and negative selection and avoid to repeat protocol.
· Page 3, Line 36:
Please detail about Aptamer Synthesis
· Page 4, line 6:
Please give details about LC-MS/MS analysis. These details provide a brief overview of the key components and considerations when using LC-MS/MS for analytical purposes. The specific parameters and equipment used can vary depending on the nature of the samples and the analytical goals of the experiment.
· Page 4, line 13:
Please specify which reference
· Page 4, line 20-24:
Please give details about the assays. Specific details of an assay can vary significantly depending on its purpose, the analyte, and the scientific field in which it is applied. Providing clear and comprehensive information about each of these aspects is essential for reproducibility and understanding the assay's validity and reliability.
· Page 4, 2.1 and 2.2
Material and equipped not in list, please refer to other articles of Methods and Procedures.
· Page 5, Figure 1:
Some improvements in legend. E.G. Sample tube, only write biomarkers (for Shure they are expected).
· Please specify in text, full experimental details
· In procedure, the English should be improved, e. g. send samples.
· Enhancements are needed for the expected results, and it is advisable to include additional conclusions pertaining to both the methodology and the anticipated outcomes.
Comments on the Quality of English LanguageAbove
Author Response
Manuscript ID: mps-2657897
REPLY TO REVIEWER 3
Reviewer(s)' Comments to Author:
Reviewer: 3
Comments and Suggestions for Authors
I am submitting my revision of the Manuscript ID mps-2657897, entitled “ Bio-SELEX: A strategy for biomarkers isolation directly from 2 biological samples”.
This manuscript presents a novel method, Bio-SELEX, for the discovery of new biomarkers within biological samples. The concept explored in this paper is interesting and merits further investigation. The content aligns with the Journal's scope.
Nevertheless, there are several areas that require improvement before it can be published. Many crucial details are omitted, and the information provided lacks depth. Furthermore, a thorough revision of the procedural section is needed to enhance clarity and comprehensibility.
In general the experimental design need to be reviewed and English improved.
Here are some suggestions for the authors to consider during the revision process:
- Page 3, line 1 to 20:
Kindly elucidate the distinctions between positive and negative selection and avoid to repeat protocol.
Response: Thanks for the comment. We have marked with lines below the text those differential aspects between positive and negative selection to make it easier for the reader to identify. The new version is on page 3, lines 25 to 44.
- Page 3, Line 36:
Please detail about Aptamer Synthesis
Response: After the next generation sequencing (NGS), we can identify those sequences that were part of the variable region and that gave diversity to the library previously used.
5'-(Constant region)-variable region (40 nt) -(constant region)-3'.
Through bioinformatic analysis, constant regions were eliminated and variable regions (40 nt) with the highest number of reads and reads per million (RPM) were selected.
Therefore, synthesized aptamers have a length of 40nt on a scale of 200 nmoles, and purified by Polyacrylamide gel electrophoresis (PAGE), and are coupled to biotin at the 5' end.
In the new version of the manuscript, we have added some of this information that is highlighted in blue and that can be found on page 4 lines 12 to 14.
- Page 4, line 6:
Please give details about LC-MS/MS analysis. These details provide a brief overview of the key components and considerations when using LC-MS/MS for analytical purposes. The specific parameters and equipment used can vary depending on the nature of the samples and the analytical goals of the experiment.
Response: Data from used equipment [Nanoflow UPLC: Ultimate 3000 nano UHPLC system (ThermoFisher Scientific, USA)] for services contracted in the creative proteomics company, was added in the indicated section highlighted in blue which is now on page 4, line 36.
- Page 4, line 13:
Please specify which reference
Response: Thank you, it was not a reference, it was the software version that was missing and it was added and highlighted in blue on page 4 line 42.
- Page 4, line 20-24:
Please give details about the assays. Specific details of an assay can vary significantly depending on its purpose, the analyte, and the scientific field in which it is applied. Providing clear and comprehensive information about each of these aspects is essential for reproducibility and understanding the assay's validity and reliability.
Response: Thanks for the comment, the experimental validation is an additional step to the Bio-SELEX strategy that must be carried out in each laboratory depending on capacities and resources. This section mentions some techniques that can be used, but it is not explained one by one because, if you review paragraph 3 (Procedure), the Bio-SELEX covers the traditional SELEX, Pull-Down to isolate potential biomarkers and identify them by LC-MS/MS. The validation corresponds to the final application to be carried out, both the biomarker potential and the aptamer. We consider that this is not the objective of this publication, since covering all possible validations for potential applications of both infectious and non-infectious diseases would lose the focus of this Protocol.
- Page 4, 2.1 and 2.2
Material and equipped not in list, please refer to other articles of Methods and Procedures.
Response: Dear reviewer, I assume you’re referring to other articles in the journal "Methods and Protocols". In the instruction guide for authors, specifically for the publication type "Protocol" it is literally mentioned that “any materials and equipment used should be explicitly listed” (https://www.mdpi.com/journal/mps/instructions) and recent "protocol" publications, incorporate these lists (https://www.mdpi.com/2409-9279/6/5/94)
- Page 5, Figure 1:
Some improvements in legend. E.G. Sample tube, only write biomarkers (for Shure they are expected).
Response: Dear reviewer, I do not understand very well what it refers to, neither in the figure nor in the legend is "sample tube". If it refers to a "test sample", we believe it is necessary to represent that, from the same initial sample used for protein extraction used in SELEX, the biomarker can be isolated later by means of Pull-Down.
- Please specify in text, full experimental details
Response: We believe that in paragraph 3, entitled "Procedure", each of the steps described in the figure is sufficiently detailed. It is important to note that the structure of a "Protocol" publication is different from that of an "experimental article."
- In procedure, the English should be improved, e. g. send samples.
Response: Thanks for the comment, we have carefully reviewed it, and several modifications have been made and are now highlighted in blue.
- Enhancements are needed for the expected results, and it is advisable to include additional conclusions pertaining to both the methodology and the anticipated outcomes.
Response: In our opinion, the most frequent and general problems that can happen in our opinion have been treated, within each experiment should be evaluated situations that have to do with the type of sample and the desired final application.
Round 2
Reviewer 3 Report
Comments and Suggestions for Authors
Dear Authors,
I am submitting my revision 01V of the Manuscript ID mps-2657897, entitled “ Bio-SELEX: A strategy for biomarkers isolation directly from 2 biological samples”.
This manuscript presents a novel method, Bio-SELEX, for the discovery of new biomarkers within biological samples.
The work is properly presented and discussed. Is original and within the scope of the Journal. After the first revision, the authors have made the corrections suggested by the reviewers. I have no more comments and I consider that the paper is ready for publication.